# ToMiE: Towards Modular Growth in Enhanced SMPL for 3D Human Gaussians with Animatable Garments

## Abstract

In this paper, we highlight a critical yet often overlooked factor in most 3D human tasks, namely modeling humans with complex garments. It is known that the parameterized formulation of SMPL is able to fit human skin; while complex garments, *e.g.*, hand-held objects and loose-fitting garments, are difficult to get modeled within the unified framework, since their movements are usually decoupled with the human body. To enhance the capability of SMPL skeleton in response to this situation, we propose a modular growth strategy that enables the joint tree of the skeleton to expand adaptively. Specifically, our method, called ToMiE, consists of parent joints localization and external joints optimization. For parent joints localization, we employ a gradient-based approach guided by both LBS blending weights and motion kernels. Once the external joints are obtained, we proceed to optimize their transformations in $SE(3)$ across different frames, enabling rendering and explicit animation. ToMiE manages to outperform other methods across various cases with garments, not only in rendering quality but also by offering free animation of grown joints, thereby enhancing the expressive ability of SMPL skeleton for a broader range of applications.

## 1 Introduction

3D human reconstruction endeavors to model high-fidelity digital avatars based on real-world characters for virtual rendering and animating, which has been of long-term research value in areas such as gaming, virtual reality (VR), and beyond. Traditional methods, such as SMPL (Loper et al., 2015), achieve human body parameterization by performing principal component analysis (PCA) on large sets of 3D scanned meshes, allowing for the fitting of a specified identity. Recent neural rendering techniques have enabled implicit digital human modeling guided by Linear Blend Skinning (LBS) and SMPL skeleton, realizing lifelike rendering and animating from video inputs.

The neural-based 3D human rendering has been empowered by the cutting-edge technique, 3D Gaussian Splatting (3DGS) (Kerbl et al., 2023), for its real-time and high-quality novel view synthesis performance. By representing 3D gaussians under the canonical T-pose and utilizing the pre-extracted SMPL skeleton in the observation space, 3D human rendering results can be obtained from novel views in any frame. This stream of approaches proves high quality in rendering 3D humans that conform to the SMPL paradigm (*e.g.*, avatars in tight-fitting clothing). However, we raise concerns regarding its ability to handle human modeling involving daily garments, such as skirts or hand-held objects.

In Fig. 1, we show two cases to illustrate the limitations of current 3D human methods in modeling daily garments. On the one hand, characters shot in-the-wild are dressed in garments with high complexity in dynamics, rather than the tight-fitting garments configured under strict experimental conditions. These loose-fitting garments break the existing methods' paradigm by assuming that garments should be bounded to the motion warping of SMPL skeleton in the same way as the human body is. On the other hand, the movements of hand-held objects, *e.g.*, mobile phones, are highly decoupled from the human body and thus cannot be represented by the current SMPL. Rooted in the aforementioned scenarios, SMPL is observed to suffer from appearance ambiguity when attempting to fit such 3D human models.

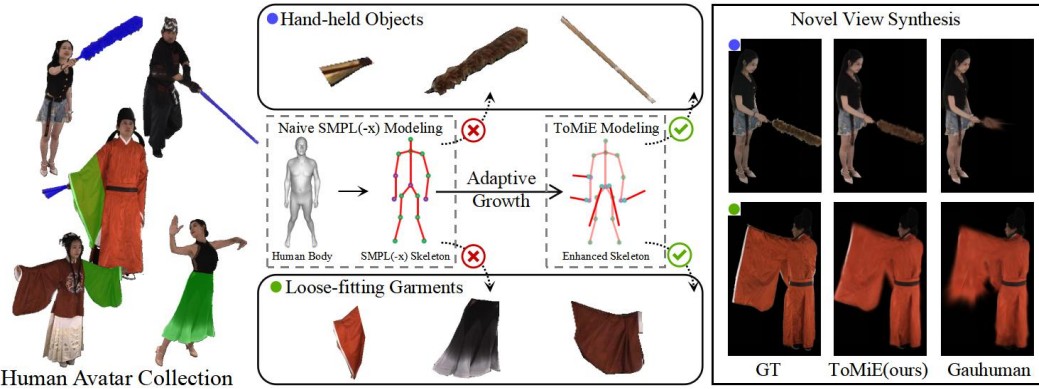

Figure 1: We show two common types of complex garments in everyday life: hand-held objects and loose-fitting garments, which cannot be accurately represented by the standard SMPL model. Our ToMiE can realize adaptive growth to enhance the representation capability of SMPL without needing for time-consuming case-specific customization, achieving state-of-the-art results in both rendering and human (including complex garments) animation.

To this end, we break through the limitations of modeling 3D human gaussians with complex garments by maintaining an extended joint tree from SMPL skeleton. Although existing SMPL model has the potential capability to manually customize additional skeleton information, time-consuming and case-by-case adjustments are necessary in this case. To overcome this issue, we extend the SMPL skeleton with additional joints for each individual case adaptively. The growth is performed in an explicit and modular manner, and enables the fitting of complex garments, offering high-quality rendering of more complex cases compared to other methods (*e.g.*, novel view synthesis results in Fig. 1).

The main challenge of extending the SMPL skeleton is to determine where and how to grow additional joints. To avoid the unnecessary memory consumption and potential overfitting caused by an arbitrary growth, we first determine which of the joints are supposed to serve as parent joints by a localization strategy. We have empirically observed that parent joints requiring growth wittness larger backpropagation gradients in their associated gaussians due to underfitting. However, determining the association of gaussians with different joints is non-trivial, as the SMPL's LBS blending weights may not work with the human with garments. To more precisely define such association, we introduce the concept of *Motion Kernels* based on rigid body priors and combine them with LBS weights, resulting in more accurate gradient-based localization. After growth, we adaptively maintain an extended modular joint tree and update the extra joints by optimizing two MLP decoders for joints positions and rotations. The proposed method, termed as ToMiE, allows for explicit animation of complex garments represented by external joints.

By experiments on complex cases with garments of the DNA-Rendering dataset (Cheng et al., 2023), ToMiE exhibits state-of-the-art rendering quality while maintaining the animatability that is significant for downstream productions. To summarize, our contributions are three-fold as follows:

1) ToMiE, a method for creating an enhanced SMPL joint tree via a modular growth strategy, which is able to decouple complex garments from the human body, thereby achieving state-of-the-art results in both rendering and explicit animation on target cases;
2) a hybrid assignment strategy for gaussians utilizing LBS weights and *Motion Kernels*, combined with gradient-driven parent joints localization, to guide the growth of external joints;
3) a joints optimization approach fitting local rotations across different frames while sharing joints positions.

## 2 RELATED WORK

### 2.1 SMPL-BASED HUMAN MESH AVATARS

Most of the recent success in digital human modeling can be attributed to the contributions of the SMPL (Loper et al., 2015) series, which parameterizes the human body as individual shape components and motion-related human poses through 3D mesh scanning and PCA. The *pose blend shapes* in SMPL describe human body deformations as a linear weighted blending of different joint poses,

significantly improving the efficiency of editing and animating digital humans. Furthermore, it has been widely adopted for human body animation, thanks to the methods (Dong et al., 2021; Shuai et al., 2022) of estimating SMPL parameters from 2D inputs. Despite their wide range of applications, SMPL and its family still suffer from inherent limitations. Since the originally scanned 3D meshes are skin-tight, according to which the pose blend shapes are learned, the model is unable to handle significantly outlying meshes, such as human with garments like skirts and hand-held objects.

## 2.2 NEURAL REPRESENTATION FOR 3D HUMAN

Methods based on neural representations, such as NeRF (Mildenhall et al., 2020) and 3DGS (Kerbl et al., 2023), have also been playing an important part in digital human reconstruction for their high-quality rendering capabilities. Early NeRF-based methods (Weng et al., 2022; Peng et al., 2021b; Kwon et al., 2021; Chen et al., 2024; Goel et al., 2023; Chen et al., 2021a; 2023; 2021b; Gafni et al., 2021; Gao et al., 2023; Geng et al., 2023) aim to reconstruct human avatars by inputting monocular or multi-view synchronized videos. Wang et al. (2022) enforce smooth priors based on neural Signed Distance Function (SDF) to obtain more accurate human geometry. Recent breakthroughs (Li et al., 2024b; Liu et al., 2023; Zielonka et al., 2023; Li et al., 2023; Hu et al., 2024b; Qian et al., 2024; Lei et al., 2024; Hu et al., 2024c; Pang et al., 2024; Jung et al., 2023; Hu et al., 2024a; Li et al., 2024a; Liu et al., 2024; Zheng et al., 2024a; Kocabas et al., 2024; Moreau et al., 2024; Jena et al., 2023; Zheng et al., 2024b) rely on 3DGS, enabling faster and more accurate rendering. All these methods register the T-pose in a canonical space and use LBS weights to guide the rigid transformations. The animation of human body can be achieved by warping points in the canonical space into the observation space that is controlled by per-frame human poses.

## 2.3 GARMENTS RENDERING AND EDITING IN DIGITAL HUMAN

We revisit the methods that consider implicitly improving garment topology. Animatable NeRF (Peng et al., 2021a) defines a per-frame latent code to capture appearance variations across each frame. Lei et al. (2024); Guo et al. (2024) additionally register global latent bones to compensate for the limitations in garments rendering. They fail to explicitly decouple the garments from human body, making precise control infeasible. Another stream (Chen et al., 2024; Hu et al., 2024c) leverages a human poses sequence as contexts to resolve appearance ambiguities. But correspondingly, they require a sequence of human poses for animation, adding to the challenges of editing. Moreover, these methods struggle to fit the object-level pose independent of the human poses sequence, such as hand-held items. We also note that some works (Hu, 2024; Men et al., 2024) introduce diffusion-based generative methods to enhance the realism of garment rendering, but these methods are restricted by the traditional SMPL and overlook the editing of complex garments. Our approach enables explicit decoupling of garments and the human body by extending the SMPL joint tree, allowing for high-quality rendering and explicit animating of garments, including hand-held objects.

## 3 PRELIMINARIES

### 3.1 SMPL(-X) REVISITED

Pre-trained on scanned meshes, the SMPL(-X) family (Loper et al., 2015; Pavlakos et al., 2019) employs a parameterized model to fit human bodies of different shapes and under different poses. The human mesh in each frame evolves from a canonical human mesh and is controlled by shape and pose parameters. Specifically, a 3D point $\boldsymbol{x}_c$ on the canonical mesh will be warped to obtain point in the observation space as

$$\boldsymbol{x}_o = \sum_{k=1}^{K} \omega_k(\boldsymbol{x}_c)(R_k(\boldsymbol{r}^0)\boldsymbol{x}_c + t_k(\boldsymbol{j}^0, \beta)), \tag{1}$$

where $K$ is the total number of joints, $R_k$ is per-joint global rotation controlled by local joint rotations $\boldsymbol{r}^0$, and $t_k$ is per-joint translation controlled by joint positions $\boldsymbol{j}^0$ and human shape $\beta$. Notice that linear blending weight $\omega_k$ is a function of $\boldsymbol{x}_c$, which is regressed from large human assets.

The main issue with this LBS-based prior model is that it can only fit tight-fitting avatars conforming to the SMPL(-X) paradigm, making it unsuitable for modeling complex human clothing in more generic cases. It is even more challenging is its inability to handle hand-held objects that are fully decoupled from the human pose. To leverage SMPL(-X) prior without being constrained by garment types, we highlight its extension to complex garment modeling.

## 3.2 HUMAN GAUSSIANS REVISITED

Human gaussians achieve high-quality real-time human rendering by combining the SMPL prior with 3DGS as the representation. The SMPL model naturally obtains the T-pose (*i.e.*, all human poses are identity transformations) mesh in the canonical space and then mesh vertices are used to initialize the canonical gaussian units. Each gaussian is defined as

$$G(\boldsymbol{x}) = \frac{1}{(2\pi)^{\frac{3}{2}}|\boldsymbol{\Sigma}|^{\frac{1}{2}}} e^{-\frac{1}{2}(\boldsymbol{x}-\boldsymbol{\mu})^T \boldsymbol{\Sigma}^{-1}(\boldsymbol{x}-\boldsymbol{\mu})}, \tag{2}$$

where $\boldsymbol{\mu}$ is the 3D gaussian center, and $\boldsymbol{\Sigma}$ is the 3D covariance matrix, which will be further decomposed into learnable rotation $\boldsymbol{R}$ and scale $\boldsymbol{S}$. Now we have $\boldsymbol{\Sigma} = \boldsymbol{R}\boldsymbol{S}\boldsymbol{S}^\top\boldsymbol{R}^\top$, which is performed by optimizing a quaternion $\boldsymbol{r}_g$ for rotation and a 3D vector $\boldsymbol{s}_g$ for scaling . Each gaussian is further assigned with color $c$ and opacity $\alpha$.

Once the 3D gaussians in the canonical space are obtained, each position $\boldsymbol{\mu}$ will be warped to the observation space according to Eq. (1). Next, the 3D gaussians of each frame are projected into 2D gaussians, followed by tile-based rasterization. Color of each pixel can be calculated by blending $N$ ordered gaussians following

$$C = \sum_{i \in N} c_i \alpha_i \prod_{j=1}^{i-1}(1 - \alpha_i). \tag{3}$$

The 3D gaussians can be optimized and updated by adaptive density control, which primarily includes cloning, splitting, and pruning. Cloning and splitting are guided by gradients to control the number of gaussians, while pruning removes empty gaussians based on current opacity $\alpha$. The supervision upon human gaussians is derived from multi-view or monocular videos, enabling high-quality rendering and avatar animation.

## 4 METHODS

Figure 2 illustrates ToMiE's modular joint growth and gaussian training strategy. Our goal is to extend the SMPL skeleton to handle complex human garments. However, the abuse of growth can lead to unnecessary computational and memory overheads. For an efficient adaptive growth, we first propose a localization strategy of parent joints to ensure that only necessary joints are grown. Furthermore, we explicitly define the hand-held joints in the $SE(3)$ space and optimize them end-to-end through an MLP, ensuring alignment with the original skeleton of SMPL. The extended skeleton can thus support rendering and explicit animation. To address the limitation of LBS in guiding gaussian attributes apart from the positions, we further fine-tune the rotation and scale during training with a deformation field to achieve better non-rigid warping. Next, we will elaborate on these modules and the training strategy in detail.

### 4.1 MOTION KERNELS-GUIDED JOINT GRADIENT ACCUMULATION

A quantitative metric needs to be identified to determine whether a joint requires growth. We notice that, due to the poor fitting ability of existing SMPL-based human gaussian, it will leave larger gradients in human regions with complex garments (*e.g.*, the hand-held object in Fig. 2 ③). In other words, *identifying joints with larger accumulated gradients can help to indicate which parent joints are more likely to require growth*. Bounded with each gaussian, the accumulated gradients will be first assigned to their corresponding joints. Let $g = \|(g_x, g_y, g_z)\|_2$ represent the L2 norm of the gradient at each gaussian position. The gradients accumulation $g_J$ for the $k$-th joint can then be computed according to

$$g_{J_k} = \frac{\sum_{i \in N} \omega_k(\boldsymbol{x}_c)g}{\sum_{i \in N} \omega_k(\boldsymbol{x}_c)}. \tag{4}$$

Figure 2: The pipeline of ToMiE. ① We initialize the gaussians in the canonical space with a standard SMPL vertices. ② (Sec. 4.4) We apply Linear Blend Skinning (LBS) to the gaussian position and utilize a network for rotation and scale correction. During the warmup phase, Adaptive LBS only utilizes the original SMPL skeleton. After adaptive growth, it further includes the newly grown external skeleton. ③ Gaussian rasterization and gradients backpropagation. ④ (Sec. 4.1, Sec. 4.2) We employ a gradient-based parent joints localization method, and a motion kernel to optimize the gradient assignment process. ⑤ (Sec. 4.3) We maintain an extra joint book with MLPs, which generates explicit human pose, enabling the decoupling and explicit animating of complex garments.

We use $\boldsymbol{x}_c$ to stand for gaussian position in canonical space and $\boldsymbol{x}_o$ in the observation space.

It is worth emphasizing that $\omega_k$ here is a weight term determining the assignment of a gaussian to the $k$-th joint. Under the paradigm of SMPL representation, this weight corresponds to the LBS weight $\omega_{\mathrm{lbs}0}$, and guides the rigid transformation of vertices on the SMPL mesh according to the human pose. To account for the differences between the human mesh with garments and the vanilla SMPL mesh, Hu et al. (2024b) calculate $\omega_{\mathrm{lbs}}$ by keeping using the LBS weight prior $\omega_{\mathrm{lbs}0}$ and adding an extra learnable network $\Phi_{\mathrm{lbs}}$ for fine-tuning. This formulation (with index $k$ omitted) can be summarized as

$$\omega_{\mathrm{lbs}}(\boldsymbol{x}_c) = \omega_{\mathrm{lbs}0}(\mathrm{NN}_1(\boldsymbol{x}_c, \boldsymbol{V})) + \Phi_{\mathrm{lbs}}(\boldsymbol{x}_c), \tag{5}$$

where $\mathrm{NN}_1$ stands for top-1 nearest-neighbor search algorithm and $\boldsymbol{V}$ represents the canonical standard SMPL vertices. It will be, however, clarified by us, that this nearest neighbor-based assigning method will no longer be feasible in cases with complex garments.

In Fig. 3, we present an example of misclassification by the nearest-neighbor search algorithm in Eq. (5). Due to the lack of human topology constraints in the canonical space, incorrect classification of hand-held objects can occur, as shown in Fig. 3 (a). This part, following the naive nearest-neighbor search, would be assigned to the leg by mistake. Fig. 3 (b) shows the points belonging to the hand joint (the correct parent joint where the handheld object should grow). Yet being solely guided by LBS weights results in the voids by misclassification.

To mitigate this issue, we propose a more robust assignment method based on motion priors, which we call *Motion Kernels*. Specifically, in the observation space, the motion kernel of each point $\boldsymbol{x}_o$ with respect to the $k$-th joint position $\boldsymbol{j}_k$ is defined based on the changes in their pairwise Euclidean distances through all input $N$ frames, following

$$\mathrm{MK}(\boldsymbol{x}_c, \boldsymbol{j}_k) = \frac{1}{N} \sum_{i=1}^{N} \left( \|\boldsymbol{x}_{oi} - \boldsymbol{j}_{ki}\|_2 - \mu \right)^2, \tag{6}$$

(a)                                                    (b)

Figure 3: Principle of the Motion Kernel. Relying solely on LBS weights will bring the misclassification in the canonical space shown in (a) to the observation space in (b), resulting in voids. Our proposed motion kernel focuses on motion-dependent priors in the observation space, offering better robustness and being less sensitive to misclassifications in the canonical space. This aids for point assignment process in parent joints localization.

and

$$\mu = \frac{1}{N} \sum_{i=1}^{N} \|\boldsymbol{x}_{oi} - \boldsymbol{j}_{k_i}\|_2 \,. \tag{7}$$

Our motion kernel (MK) reflects the relative motion between each gaussian and joint. A smaller MK indicates that the pair of gaussian and joint is relatively stationary to each other, signifying a stronger association, while a larger MK suggests a higher degree of relative motion, indicating a weaker association. We further represent the assignment weight reflected by the MK as $\omega_{\text{MK}}(\boldsymbol{x}_c) = \text{Normalize}_k(\text{MK}^{-1}(\boldsymbol{x}_c, \boldsymbol{j}_k))$, and the final assignment weight in Eq. (4) (with index k omitted) becomes

$$\omega(\boldsymbol{x}_c) = \lambda \omega_{\text{MK}}(\boldsymbol{x}_c) + (1 - \lambda)\omega_{\text{lbs}}(\boldsymbol{x}_c), \tag{8}$$

where $\lambda$ is a hyperparameter to balance the MK weight and LBS weight. Note that we do not completely abandon the LBS weight, because in practice, the MK cannot differentiate the association of points on either side of a joint, requiring the LBS weight to compensate.

## 4.2 Gradient-based Parent Joint Localization

By combining Eq. (4) and Eq. (8), the parent joints $\boldsymbol{J}_s \subseteq \boldsymbol{J}$ that require growth can be located. Basically, we have $\boldsymbol{g_J} = (g_{J_1}, g_{J_2}, \ldots, g_{J_K})$ to represent the gradient accumulation of total $K$ human joints. As mentioned in Sec. 4.1, the joints with larger gradients accumulated are more likely to require the growth of child joints. This is achieved by sorting $\boldsymbol{g_J}$ in descending order $\pi$ to get $\boldsymbol{g_J}^{\text{sorted}} = (g_{J_{\pi(1)}}, g_{J_{\pi(2)}}, \ldots, g_{J_{\pi(K)}})$.

We set a gradient threshold $\epsilon_{\boldsymbol{J}}$ to identify the $J \in \boldsymbol{J}_s$ that requires growth, following

$$\boldsymbol{J}_s = (J_{\pi(1)}, J_{\pi(2)}, \ldots, J_{\pi(N)}) \quad s.t. \quad g_{J_{\pi(N)}} \geq \epsilon_{\boldsymbol{J}} \text{ and } g_{J_{\pi(N+1)}} < \epsilon_{\boldsymbol{J}}, \tag{9}$$

where we can safely assume $N < K$.

For each joint in $\boldsymbol{J}_s$, we designate it as a parent joint requiring growth and proceed with the initialization of its child joint. In order to explicitly model each child joint and ensure the consistency with the SMPL paradigm for ease of animation, we maintain an extra joint book $B^e = (\text{parent}, \boldsymbol{j}^e, \boldsymbol{r}^e)$, that includes the indices of parent joints, the extra joint positions $\boldsymbol{j}^e$ in canonical space and the extra rotations $\boldsymbol{r}^e$ in its parent joint coordinate. We initialize the joint position to its parent joint's position and set the rotation to the identity rotation. The entire parent joint localization and child joint initialization process is guided by the gradients, effectively preventing unnecessary overgrowth and ensuring a dense distribution of the extra joints.

## 4.3 Extra Joint Optimization

The joint positions and rotations in the extra joint book are set as optimizable, which will be stored and later decoded by two shallow MLPs. According to the SMPL paradigm, the canonical joint

position is a time-invariant quantity, therefore the joint position optimization network $\Phi_p$ is defined as

$$dj^e(i) = \Phi_p(P.E.(i)), \tag{10}$$

where *P.E.* is a positional encoding function as is in (Mildenhall et al., 2020) and $i$ is the joint index in $B^e$. Rotations are also dependent on the timestamp $t$ of each frame, thus the rotation optimization network $\Phi_r$ is defined as

$$r^e(i, t) = \Phi_r(P.E.(i), P.E.(t)). \tag{11}$$

Now we have the positions of extra joints $j^e$ (optimized by the offset $dj^e$) and the rotations $r^e$. Each extra joint position is defined in the canonical space, representing the intrinsic properties of the extended skeleton, while the extra joint rotation in parent joint coordinate varies frame by frame.

Although both the extra joint positions and rotations are stored in the MLPs, their inputs and outputs are explicit features with real physical meaning, which allows for both implicit and explicit editing. For example, we can interpolate over timestamp $t$ based on $\Phi_r$ or directly use explicit inputs to take the place of $\Phi_r$ during animation. The MLPs here function as decoders, deriving joint-related values from indices and timestamps, thus effectively circumventing the need for explicit storage of joint values, as employed in SMPL.

### 4.4 Inference Process and Training Strategy

In this subsection, we explain how our modular growth method is integrated with the training process of human gaussians.

First, we initialize the canonical gaussians with standard SMPL vertices. At the beginning of training, we set up a number of warm-up iterations during which no joint growth occurs, and the gaussian fitting is performed following the traditional human gaussian methods. This prevents underfitting due to insufficient training, which could further affect the joint localization in Sec. 4.2.

During the warm-up iterations, canonical human gaussians will be first warped to the observation space according to the LBS weight in Eq. (5). To compensate for the inability of the LBS model to represent gaussians rotation and scale, a deformable network $\Phi_d$ is employed to correct the rotation and scale during the LBS process. To distinguish it from the joint rotation $r$ of the SMPL human pose, we denote the rotation of the gaussian with subscript $g$, and

$$dr_g, ds_g = \Phi_d(x_g, r^0). \tag{12}$$

The gaussians rotation $r_g$ and scale $s_g$ are modified with offsets $dr_g$ and $ds_g$ to get the final gaussians in the observation space. In the observation space, we obtain the rendered images through the rasterisation of gaussians and compute the image loss to supervise canonical gaussians.

Once the warm-up iterations completes, we begin the modular joint growth. With the MK calculated during the warm-up phase, we can locate the parent joints $J_s$ that require growth. Then we add grown joints to the extra joint book $B^e$, optimizing its positions $j^e$ and rotations $r^e$ during the subsequent training. Notably, $\Phi_{lbs}$ needs to extend its output dimensions to include the blending weights for extra joints, following $K = K^0 + K^e$. Since the extra joints do not have prior LBS weights, their blending weights are entirely learned through $\Phi_{lbs}$.

Both the warm-up stage and the post-growth learning stage adopt adaptive density control to manage the updates of the gaussian points. We also dynamically adjust the threshold of gradient for densification in (Kerbl et al., 2023) based on the number of gaussians, in order to balance memory consumption. Please check our supplemental material for details of this design.

## 5 Experiments

### 5.1 Dataset

Our method focuses on complex garments and hand-held objects, so we select datasets for experiments accordingly. We notice that the DNA-Rendering dataset (Cheng et al., 2023), by capturing complex scenes of the human body, meets our requirements. Specifically, we selected 8 cases that align with our hypothesis, namely *0041_10, 0090_06, 0176_07, 0800_07* (hand-held objects) and

Table 1: Quantitative comparison between our method and other runners-up. $\mathbb{B}$ and $\mathbb{G}$ stand for human body and complex garments. We color each result as best , second best and third best .

| Dataset Method \| Metric | Animatable | | DNA-Rendering | | | | ZJU-Mocap | | |
|---|---|---|---|---|---|---|---|---|---|
| | $\mathbb{B}$ | $\mathbb{G}$ | PSNR(full)↑ | SSIM↑ | LPIPS↓ | PSNR(masked)↑ | PSNR(full)↑ | SSIM↑ | LPIPS↓ |
| 3DGS-Avatar | ✓ | ✗ | 27.69 | 0.948 | 0.0496 | 16.96 | 30.61 | 0.970 | 0.0296 |
| GART | ✓ | ✓ | 29.25 | 0.958 | 0.0480 | 17.12 | 30.91 | 0.962 | 0.0318 |
| Im4D | ✗ | ✗ | 26.28 | 0.965 | 0.0308 | 14.90 | 28.99 | 0.973 | 0.0620 |
| GauHuman | ✓ | ✗ | 30.22 | 0.962 | 0.0405 | 18.26 | 30.82 | 0.962 | 0.0326 |
| ToMiE(ours) | ✓ | ✓ | 31.28 | 0.966 | 0.0374 | 19.75 | 31.10 | 0.963 | 0.0312 |

*0007_04, 0007_04, 0051_09, 0811_06* (loose-fitting garments). For each case, we use 24 surrounding views for training and 6 novel surrounding views for testing. All views are synchronized and contained 100 frames each.

In addition to tackle complex garments, it is essential to ensure the model's performance in typical scenarios involving tight-fitting clothes. Therefore, we additionally test our method on the ZJU-MoCap (Peng et al., 2021b) dataset. Although the tight-clothing cases are too simple to require joint extension, our overall framework still achieves optimal results. Since this part of the experiment is not directly related to modular growth, we refer the readers to the supplemental materials for further visualizations.

## 5.2 BASELINES AND METRICS

Given the wide variety of work on the human body, we select only the most cutting-edge and representative works from each focus area for a fair comparison. Since 3DGS is currently the leading representation for novel view synthesis, we compare 3DGS-based methods, including 3DGS-Avatar (Qian et al., 2024), GART (Lei et al., 2024), and GauHuman (Hu et al., 2024b). Among them, GART is expected to offer extra animatability for its modeling of implicit global auxiliary bones. Additionally, there is another category of human modeling without incorporating SMPL-like pose priors. Although these methods don't guarantee an animatable human avatar, they can achieve high-quality rendering, among which, we compare the rendering quality of Im4D (Lin et al., 2023) with our method.

We conduct a comprehensive qualitative and quantitative comparison of our ToMiE against these methods. We report three key metrics: peak signal-to-noise ratio (PSNR), structural similarity index measure (SSIM) (Wang et al., 2004), and learned perceptual image patch similarity (LPIPS) (Zhang et al., 2018). Per-scene results can be found in the supplemental material. In addition to comparing the rendering results, we also demonstrate the animatability of the extra garments enabled by our method. We strongly recommend readers to watch the supplemental video for a more intuitive understanding of the animating results.

## 5.3 NOVEL VIEW SYNTHESIS RESULTS

Figure 4 and Tab. 1 present the results of our method compared to other baselines. In the tables, we showcase two evaluation protocols. The first evaluates the entire image, reflecting the overall rendering quality. The second uses a binary mask to specifically compare the complex garments regions, demonstrating how our method outperforms others in these challenging cases. The mask is shown in Fig. 4, and details for its calculation can be found in the supplemental materials.

## 5.4 ABLATION STUDIES

**A. Modular Growth Ablation**. We remove the adaptive growth process to ablate its impact on the rendering results. Table 2 "w/o A" shows a decline in rendering quality, while the human garments also become not animatable.

**B. Non-rigid Design Ablation**. In Sec. 4.4, we apply a non-rigid deformation network $\Phi_d$ to correct gaussian rotation and scale. As shown in Tab. 2 "w/o B", this significantly improves the final rendering quality.

Table 2: Ablation studies on DNA-Rendering dataset (Cheng et al., 2023). We independently ablate the modular growth strategy and non-rigid design to validate their impact on the overall performance.

| | PSNR(full) ↑ | SSIM ↑ | LPIPS ↓ | PSNR(masked) ↑ |
|---|---|---|---|---|
| w/o Modular Growth | 30.99 | 0.964 | 0.0390 | 19.21 |
| w/o Non-rigid Design | 30.78 | 0.964 | **0.0363** | 19.38 |
| Full | **31.28** | **0.966** | 0.0374 | **19.75** |

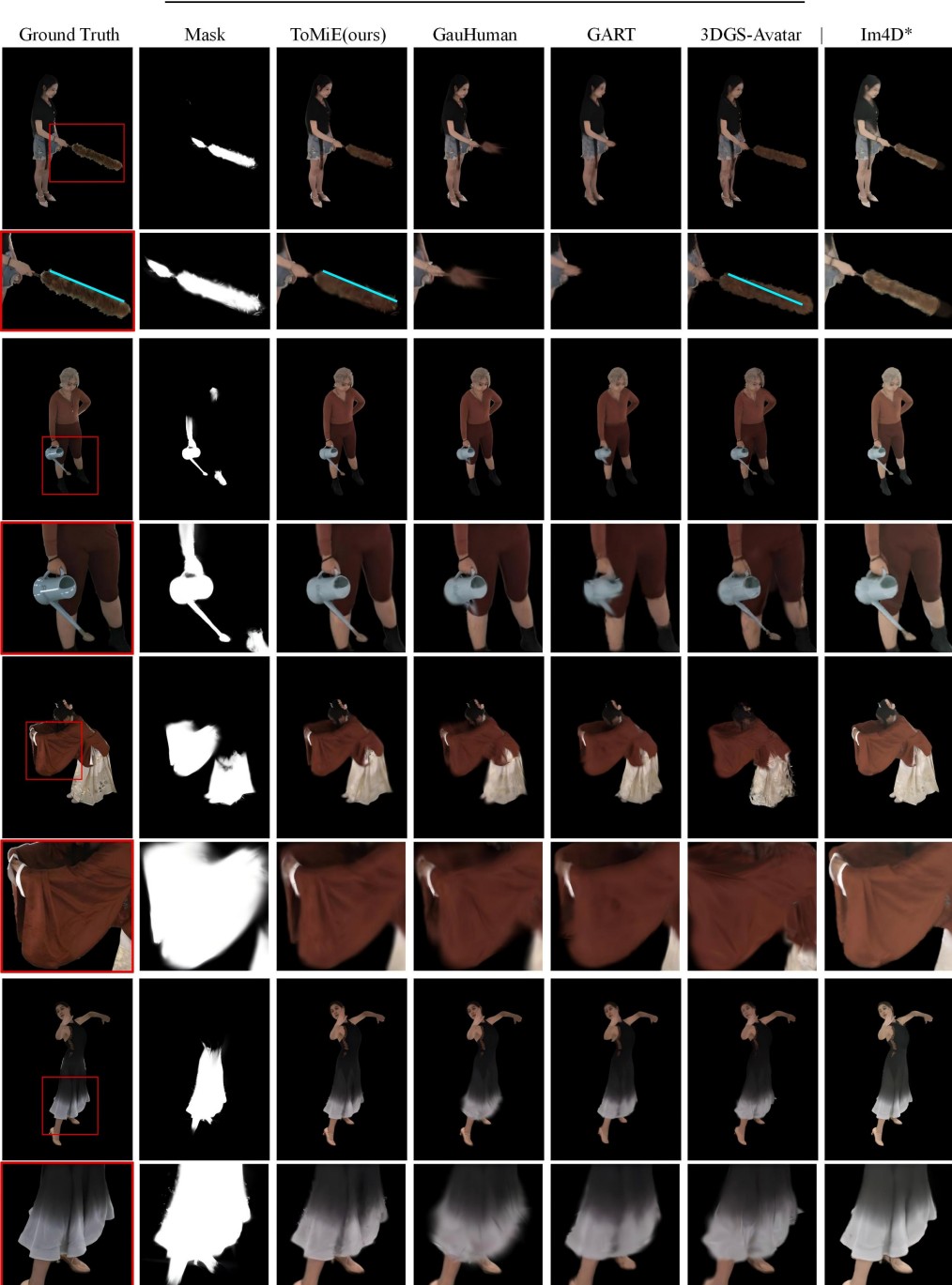

Figure 4: Qualitative comparison on the DNA-Rendering dataset (Cheng et al., 2023). We show two cases of hand-held objects (*0041_10, 0800_07*) and two cases of loose-fitting garments (*0007_04, 0811_06*) (from top to bottom). 3DGS-Avatar (Qian et al., 2024) can fit hand-held objects in some cases (*e.g.*, *0041_10*), its pose becomes incorrect due to the lack of proper joint optimization (cyan lines are parallel). Im4d* (Lin et al., 2023) achieves high-quality rendering but cannot be animated.

## 5.5 Animating Hand-held Objects and Loose-fitting Garments

Figure 5: Animating Results of ToMiE and GART (Lei et al., 2024). Our explicit modeling fully decouples garments from the human body, enabling part-specific animating. However, implicit modeling of garments conditioned on SMPL human poses leads to the failure of animating of GART.

We demonstrate the uniqueness of our method, specifically its ability to explicitly animate complex garments. Our animating approach can be implemented in two ways. On the one hand, we can utilize the transformation already recorded in the extra joint book to replay garments motions. The visualization of this part is shown in Fig. 5. On the other hand, ToMiE also supports bypassing the decoding process of the extra joint book by directly inputting the transformation explicitly. This allows us to customize the motion trajectory of external joints. Due to the limitations of the image's expressive capabilities, we strongly recommend the readers to watch the supplemental video to check the animated results under the "spiral" trajectory.

In Fig. 5, we edit the extra joints of complex garments while keeping the body poses under the SMPL paradigm stationary. Since the implicit auxiliary bones of GART (Lei et al., 2024) is controlled by the traditional SMPL poses, only the identical appearance can be output when SMPL poses are stationary. In contrast, our method explicitly models the garments, fully decoupling them from the traditional SMPL poses, enabling free animating.

## 6 Limitations and Conclusion

**Limitations**. Although our method enhances the modeling for rigid and non-rigid garments, it cannot address scenarios involving drastic changes in the topology. This is because topological changes disrupt the one-to-one correspondence between frames, making the human modeling centered on the canonical space become downgraded. We notice that Park et al. (2021) address topology issues by introducing high-dimensional mappings, which could be adapted to build our non-rigid deformation. However, this is not the main scope of this paper and can be explored as future work.

**Conclusion**. In this paper, we introduce ToMiE, a modular growth method designed to extend traditional SMPL skeleton for better modeling of human garments. In the first stage, we assign the gradient of gaussian points to different joints by combining LBS weights with the motion kernel based on motion priors. This allows us to accurately locate the parent joints that need to grow, avoiding redundant growth. In the second stage, we design an extra joint book to achieve explicit joint modeling and optimize the transformation of the newly grown joints in an end-to-end manner. With the improved designs mentioned above, our ToMiE stands out among numerous state-of-the-art methods, achieving the best rendering quality and animatability of garments. We hope the adaptive growing method will spark a renewed discussion on the current capabilities of digital human modeling. What's more, it is expected to offer some insights for subsequent works related to topology and skeleton generation.

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

# A APPENDIX

## A.1 PER-SCENE RESULTS ON DNA-RENDERING DATASET

We exhibit our per-scene results on DNA-Rendering dataset (Cheng et al., 2023) in Tab. 3. To more clearly demonstrate the results of the growth, we further present the parent joints $J \in \boldsymbol{J}_s$ for each case in Tab. 4, as defined in Sec. 4.2.

Table 3: Per-scene quantitative comparisons on DNA-Rendering dataset.

| | 0007_04 | | | | 0014_06 | | | | 0041_10 | | | |
|---|---|---|---|---|---|---|---|---|---|---|---|---|
| Method | PSNR(full) ↑ | SSIM ↑ | LPIPS ↓ | PSNR(masked) ↑ | PSNR(full) ↑ | SSIM ↑ | LPIPS ↓ | PSNR(masked) ↑ | PSNR(full) ↑ | SSIM ↑ | LPIPS ↓ | PSNR(masked) ↑ |
| 3DGS-Avatar | 23.42 | 0.928 | 0.072 | 13.92 | 25.40 | 0.926 | 0.085 | 16.09 | 27.00 | 0.934 | 0.057 | 18.33 |
| GART | 28.23 | 0.952 | 0.057 | 17.80 | 26.87 | 0.942 | 0.075 | 17.16 | 28.45 | 0.942 | 0.060 | 15.26 |
| Im4D | 25.61 | 0.953 | 0.037 | 14.86 | 26.36 | 0.955 | 0.042 | 16.41 | 25.75 | 0.946 | 0.046 | 16.51 |
| GauHuman | 28.03 | 0.948 | 0.054 | 17.85 | 27.45 | 0.941 | 0.071 | 17.84 | 29.41 | 0.946 | 0.052 | 16.14 |
| ToMiE(ours) | 29.12 | 0.953 | 0.050 | 18.88 | 28.72 | 0.947 | 0.064 | 19.46 | 30.43 | 0.950 | 0.048 | 19.69 |
| | 0051_09 | | | | 0090_06 | | | | 0176_07 | | | |
| Method | PSNR(full) ↑ | SSIM ↑ | LPIPS ↓ | PSNR(masked) ↑ | PSNR(full) ↑ | SSIM ↑ | LPIPS ↓ | PSNR(masked) ↑ | PSNR(full) ↑ | SSIM ↑ | LPIPS ↓ | PSNR(masked) ↑ |
| 3DGS-Avatar | 25.06 | 0.953 | 0.041 | 14.26 | 32.88 | 0.968 | 0.031 | 19.60 | 27.93 | 0.954 | 0.034 | 15.99 |
| GART | 26.45 | 0.963 | 0.042 | 14.92 | 34.50 | 0.975 | 0.030 | 20.43 | 29.87 | 0.963 | 0.032 | 16.38 |
| Im4D | 24.36 | 0.970 | 0.030 | 11.99 | 28.20 | 0.975 | 0.023 | 15.05 | 26.03 | 0.982 | 0.014 | 13.87 |
| GauHuman | 26.24 | 0.961 | 0.039 | 14.75 | 36.42 | 0.983 | 0.021 | 22.91 | 32.25 | 0.980 | 0.018 | 19.86 |
| ToMiE(ours) | 27.43 | 0.965 | 0.037 | 15.63 | 36.70 | 0.984 | 0.020 | 22.89 | 32.77 | 0.982 | 0.017 | 20.29 |
| | 0800_07 | | | | 0811_06 | | | | average | | | |
| Method | PSNR(full) ↑ | SSIM ↑ | LPIPS ↓ | PSNR(masked) ↑ | PSNR(full) ↑ | SSIM ↑ | LPIPS ↓ | PSNR(masked) ↑ | PSNR(full) ↑ | SSIM ↑ | LPIPS ↓ | PSNR(masked) ↑ |
| 3DGS-Avatar | 31.27 | 0.962 | 0.033 | 18.59 | 28.60 | 0.959 | 0.044 | 18.89 | 27.69 | 0.948 | 0.050 | 16.96 |
| GART | 31.94 | 0.968 | 0.035 | 17.48 | 27.68 | 0.961 | 0.052 | 17.52 | 29.25 | 0.958 | 0.048 | 17.12 |
| Im4D | 27.56 | 0.970 | 0.023 | 14.88 | 26.36 | 0.966 | 0.032 | 15.60 | 26.28 | 0.965 | 0.031 | 14.90 |
| GauHuman | 33.48 | 0.976 | 0.024 | 18.87 | 28.51 | 0.961 | 0.046 | 17.86 | 30.22 | 0.962 | 0.041 | 18.26 |
| ToMiE(ours) | 34.44 | 0.977 | 0.021 | 21.01 | 30.62 | 0.967 | 0.042 | 20.17 | 31.28 | 0.966 | 0.037 | 19.75 |

Table 4: Description of human action and index of grown parent joints $\boldsymbol{J}_s$ for each sequence. Please refer to the joint positions in Fig. 6 for a better understanding of the grown joints.

| Sequence | Description | grown parent joints $\boldsymbol{J}_s$ |
|---|---|---|
| 0007_04 | Waving sleeves | [18, 19, 20, 10, 21, 7, 8, 4, 11, 5, 16, 1] |
| 0014_06 | Waving sleeves | [10, 20, 18, 16, 21, 7, 4, 19, 1, 5, 8, 17, 11, 13] |
| 0041_10 | Swinging a feather duster | [21, 13] |
| 0051_09 | Flowing a dress | [ 8, 4, 12, 5, 1, 7, 11, 10, 2] |
| 0090_06 | Tending a bonsai | [18, 20, 16, 13] |
| 0176_07 | Using a hairdryer | [21] |
| 0800_07 | Watering with a kettle | [19, 21, 13, 10] |
| 0811_06 | Spinning a dress | [4, 7, 2, 1, 5, 0] |

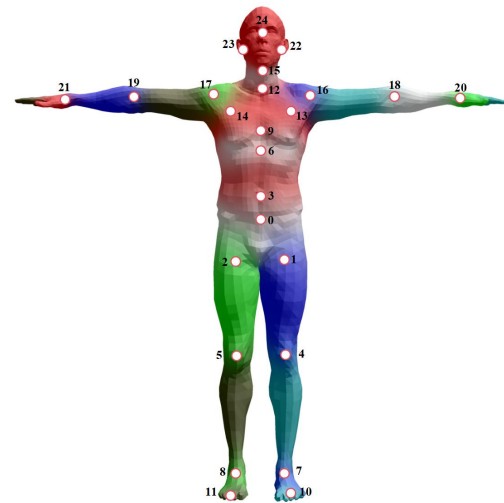

Figure 6: Joints distribution we use in our method. We use the SMPL-X model (Pavlakos et al., 2019) while removing the MANO (Romero et al., 2017) joints in hands, as we experimentally find that the MANO joints in DNA-Rendering data are inaccurately labeled. More empirically, there is no need to grow extra joints for the fingers.

## A.2 Visualization on ZJU-Mocap Dataset

In Sec. 5.1, we quantitatively experiment on ZJU-Mocap (Peng et al., 2021b) dataset to validate that our method is also effective in scenarios with tight-fitting garments. In Fig. 7, we present more visualization results.

| Ground Truth | ToMiE(ours) | GauHuman | Ground Truth | ToMiE(ours) | GauHuman |

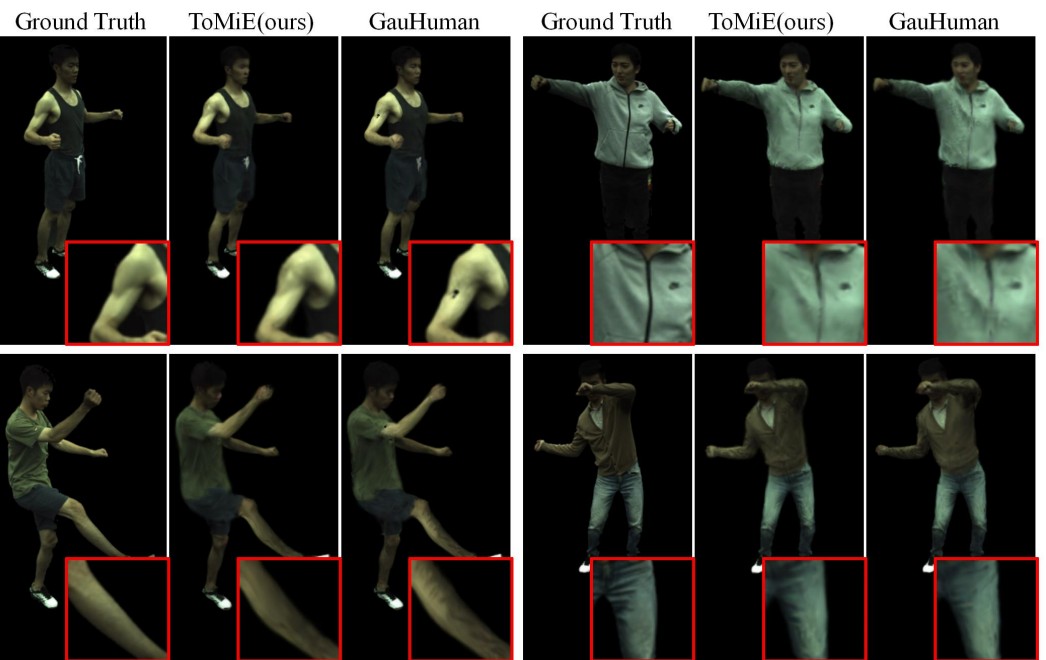

Figure 7: Qualitative results on ZJU-Mocap (Peng et al., 2021b) dataset. Please check the zoom-in areas to find that our method reconstructs more details compared to GauHuman (Hu et al., 2024b), even in tight-fitting cases where growth of extra joints is not required.

## A.3 Calculation of Masks with Complex Garments

Our method focuses primarily on modeling complex garments. In Tab. 1, we further evaluate the model's performance in these areas using a binary mask. To generate a mask for each scene that distinguishes regions containing complex garments, we first segment the potential complex garment points in 3D space. Specifically, points predominantly controlled by the extra joints and their parent joints are identified as part of the complex garments. With the pre-trained blending weight, we can easily locate these points, which are then assigned a white color, while all others are marked in black, forming a 3D binary mask. Finally, we obtain a 2D binary mask representing complex garments by applying Gaussian rasterization to the 3D mask. Since we rely solely on this binary mask for metrics evaluation, this post-processing method for calculating masks is permissible.

## A.4 Adjustment of the Gradient Threshold for Densification

In scenarios with complex garments, to prevent excessive gaussian points from causing high memory consumption, we propose an adaptive suppression strategy to keep the number of gaussian points within a reasonable range. This is achieved by dynamically adjusting the threshold of the gradient for densification $\epsilon_d$ in (Kerbl et al., 2023). This threshold $\epsilon_d$ represents that points with accumulated gradients exceeding it will be densified. Therefore, a larger threshold results in fewer split points, and vice versa.

Let us assume the desired maximum number of gaussian points is $N$. After each iteration, if the current number of gaussian points $n$ exceeds $N$, we will increase $\epsilon_d$ according to

$$\epsilon_d = (a + \frac{n - N}{b})\epsilon_{d_0}. \tag{13}$$

In the practical implementation, we set $N = 3 \times 10^4$, $a = 2$, $b = 5 \times 10^3$, and $\epsilon_{d_0} = 5 \times 10^{-4}$.

### A.5 DETAILS OF HYPERPARAMETERS

The $\lambda$ of balancing MK weight and LBS weight in Eq. (8) is set to $0.4$. The gradient threshold $\epsilon_J$ to identify the $J \in \boldsymbol{J}_s$ that requires growth in Eq. (13) is set to $3.5 \times 10^{-6}$. For network hyperparameters, we detail the number of layers and the width of the MLP network design. $\Phi_{\text{lbs}}$ has $D = 4$ and $W = 128$. $\Phi_p$ has $D = 4$ and $W = 256$. $\Phi_r$ has $D = 4$ and $W = 128$. $\Phi_d$ has $D = 2$ and $W = 128$. Specifically, we initialize the weights of the last layer in $\Phi_p$ and $\Phi_r$ as a tiny value $1 \times 10^{-2}$. This provides stability for the initial training phase. Warm-up iterations number in Sec. 4.4 is set to $8 \times 10^3$.

### A.6 SUPPLEMENTAL VIDEO

Our supplemental video consists of three parts. First, we present monocular rendering results to demonstrate that we can accurately render complex garments on the human body, including hand-held objects. Next, we perform 360-degree rendering of the full video to validate our generalization capability on novel views. Finally, we fix standard human skeleton still, only animating extra garments to show our decoupling and explicit animating capability. Since image quality may not fully demonstrate the effectiveness of human reconstruction, especially for animating results, we recommend that readers refer to this supplemental video for better visualization.

