# OpenReview forum: "ToMiE: Towards Modular Growth in Enhanced SMPL Skeleton for 3D Human Gaussians with Animatable Garments"
_ICLR.cc/2025/Conference — ICLR 2025 Conference Withdrawn Submission_

### Official Review · Reviewer_Xgt9 · 2024-11-02

**Soundness:** 3
**Presentation:** 3
**Contribution:** 3
**Rating:** 5
**Confidence:** 3

**Summary:**

This manuscript contributes to the field of 3D human reconstruction by proposing ToMiE, a method that addresses the challenges of modeling complex garments and improves the quality of both rendering and animation.  The hybrid assignment strategy and joints optimization approach further enhance the capabilities of ToMiE, making it a valuable tool for creating high-fidelity digital avatars in various applications, including gaming and virtual reality.  The contributions of this manuscript are three-fold:

* ToMiE Method: The authors propose ToMiE, a novel method that enhances the SMPL joint tree through a modular growth strategy.  By extending the SMPL skeleton with additional joints, ToMiE is able to decouple these garments from the human body, achieving plausible results in both rendering and explicit animation.

* Hybrid Assignment Strategy: The manuscript introduces a hybrid assignment strategy for Gaussians that combines LBS weights and Motion Kernels.  This strategy, along with gradient-driven parent joint localization, guides the growth of external joints.

* Joints Optimization Approach: The authors present a joints optimization approach that fits local rotations across different frames while sharing joint positions.  This method improves the overall quality of the animations and ensures that the avatars move naturally and realistically, even in complex scenarios involving garments and hand-held objects.

**Strengths:**

Based on the SMPL parametric model, this article makes targeted improvements to the modeling of complex garments and hand-held objects, addressing a practical yet rarely addressed issue.  The core idea of the article is straightforward, which is to enhance the SMPL joint tree through a modular growth strategy.  The proposed assignment strategy and joint optimization are technically sound.  The paper is clearly articulated and easy to follow.  The qualitative and quantitative experiments do support the statements and conclusions presented.  The authors further validate the effectiveness of the proposed module through an ablation study.  The ideas proposed in this article will likely inspire further community research.

**Weaknesses:**

The paper's novelty is limited but above the bar. My main concerns about this article lie in its rendering quality and experimental section.   In the qualitative comparisons presented by the authors, the proposed method appears overall blurry and lacks high-frequency details, which somewhat compromises realism.   Particularly in the "garments animating" part of the supplementary video,  the modeling of non-rigid motion for loose sleeves is particularly poor, lacking the necessary realism and fluidity.   This shortcoming should be adequately addressed and discussed in the limitations section.

Furthermore, the number of experimental examples is too small to fully and adequately validate the proposed method's effectiveness and generality. I suggest that the authors supplement more experimental examples to demonstrate the method's strengths and limitations more comprehensively.

Regarding the experimental comparisons, I noticed that only a comparison with the GauHuman method was conducted in the video, with no comparisons to other related methods.   This single comparison may not fully reflect the advantages and disadvantages of the proposed method nor facilitate an objective and comprehensive evaluation by readers.

**Questions:**

I have explained the questions and suggestions about the experimental part in the Weaknesses section. Please refer to the Weaknesses section.

Regarding the method description, specifically the introduction to the pipeline of ToMiE, I found that the main text is somewhat disconnected from the content of Figure 2.  I would suggest that the authors summarize and refine the content of Figure 2, ensuring that the module division aligns as closely as possible with the chapter division.

---

### Official Review · Reviewer_qLNy · 2024-11-03

**Soundness:** 2
**Presentation:** 3
**Contribution:** 2
**Rating:** 5
**Confidence:** 4

**Summary:**

The paper aims to tackle animations of hand-held objects and losse-fitting garments. The proposed method enhances the SMPL by introducing extra joints, which are obtained through the metric defined by the author. To better deal with hand-held objects, the author proposes “Motion Kernels” to correct the blending weight. Experiments show the improvement delivered by the proposed method.

**Strengths:**

* The proposed method can better fit scenarios with hand-held objects and loose garments.
* The proposed method delivers lower rendering errors and better qualitative results in most cases.

**Weaknesses:**

1. The improvements in both quantitative and qualitative results are limited, which is the main concern. In Table 1, the values are similar to each other, with small improvements comparing with baseline. In addition, some baselines, such as the 3DGS-Avatar in Figure 4, could achieve similar quality of rendering results comparing to ToMiE, even for cases with hand-held objects. It seems less convincing that ToMiE is better than other methods.
2. In terms of animation, since 3DGS-Avatar is also based on Gaussian representation and predicts the transformations to pose the kernels, it could be animated through the second way metioned in Section 5.5 around L511-514, which further weakens the contribution of this paper.

In conclusion, this paper could be further polished.

**Questions:**

Is the method garment-specific? Are separate models trained from scratch for different cases, such as the 4 different garments in Figure 4?

---

### Official Review · Reviewer_VTu5 · 2024-11-04

**Soundness:** 3
**Presentation:** 3
**Contribution:** 3
**Rating:** 5
**Confidence:** 4

**Summary:**

ToMiE is a novel method that extends the SMPL skeleton through a modular growth strategy to model 3D humans with complex garments and hand-held objects. Traditional SMPL-based models are effective for skin-tight clothing but struggle with loose garments or accessories that move independently of the body. ToMiE addresses this limitation by:

- Parent Joint Localization: Utilizing a gradient-based approach guided by both Linear Blend Skinning (LBS) weights and Motion Kernels to determine where the skeleton should be extended.
- External Joint Optimization: Optimizing the transformations of newly added joints across different frames, allowing for realistic rendering and explicit animation of complex garments.

Experiments on the DNA-Rendering dataset demonstrate that ToMiE outperforms existing methods in rendering quality and provides enhanced animatability for complex garments. The method allows for adaptive skeleton growth, enabling more accurate modeling of loose-fitting clothes and hand-held objects.

**Strengths:**

- Approach: The introduction of a modular growth strategy to extend the SMPL skeleton is a creative solution to model complex garments and accessories that traditional methods struggle with. Also, the use of Motion Kernels to guide parent joint localization improves the robustness of the model in assigning gaussians to joints, addressing issues with misclassification in previous approaches.
- Ability to animate: By optimizing external joints, ToMiE allows for explicit animation of complex garments, offering greater control and flexibility in applications like virtual reality and gaming.
- Experiments: The paper provides thorough experimental validation on the DNA-Rendering dataset, including comparisons with state-of-the-art methods and ablation studies that highlight the effectiveness of each component.
- Clarity: The paper is well-structured and clearly explains the methodology, with helpful figures that aid in understanding complex concepts.

**Weaknesses:**

- Incomplete Discussion of Related Work: The paper overlooks several relevant works in modeling complex garments, such as Multi-Garment Net (MGN), SMPLicit, CAPE, and HumanCoser. Including these in the related work section would better contextualize ToMiE within the existing literature and clarify its unique contributions.
- Limited Dataset Evaluation: The experimental validation is primarily conducted on the DNA-Rendering dataset. This limited evaluation may not fully show the generalizability of ToMiE to other datasets with diverse garments and accessories.
- Lack of Computational Cost Analysis: The adaptive skeleton growth strategy could introduce additional computational and memory overhead. The paper does not provide quantitative analysis of training and inference times or memory consumption compared to other methods, which is important for practical applications.
- Hyperparameter Sensitivity: The method involves several hyperparameters (e.g., gradient thresholds, balancing factors), but lacks a sensitivity analysis. Without guidelines on setting these parameters, it may be challenging for others to reproduce the results or apply the method to different datasets.
- Handling Drastic Topological Changes: While ToMiE improves the modeling of complex garments, it does not address scenarios involving significant topological changes, such as garments being added or removed between frames. A discussion on potential solutions or future work in this area would enhance the paper.

**Questions:**

- As mentioned above, could you please compare and/or provide justification on how ToMiE compares with existing methods like MGN, SMPLicit, CAPE, and HumanCoser? What are the key differences in handling complex garments, and why were these works not cited in your paper?
- Have you tested ToMiE on in-the-wild datasets or other datasets with different types of garments and accessories? How does your method perform in those scenarios, and does it generalize well?
- Can you provide more details on the computational costs associated with the modular growth strategy? Specifically, how does the number of extra joints affect training and inference times, and what are the memory implications?
- Have you conducted experiments to assess the sensitivity of your method to hyperparameters like \lambda and \epsilon_J? Any guidelines or recommendations for setting these values to achieve optimal performance?
- Do you have plans to extend ToMiE to handle drastic topological changes, such as when garments are added or removed between frames? How might your approach be adapted to address this limitation?

**Details Of Ethics Concerns:**

N/A.

---

### Official Review · Reviewer_vVhu · 2024-11-07

**Soundness:** 2
**Presentation:** 2
**Contribution:** 2
**Rating:** 3
**Confidence:** 5

**Summary:**

This paper introduced a modular growth method to extend original SMPL skeleton for better modeling of human garments. Motion kernels are used for motion priors to locate the parent joints. A joint book approach is also proposed to jointly model] and transformation of newly grown joints. Experiments are shown to validate the proposed method. However, there are some questions that remains. Please see the weakness section.

**Strengths:**

Please see the weakness section.

**Weaknesses:**

- 136-137 the SMPLICIT CVPR 2021 also considered the loose garments along with other accessories. Please include and discuss this paper.
- Author must discuss the technical similarity/dissimilarity with virtual bones paper Predicting Loose-Fitting Garment Deformations in the context of extending SMPL skeleton.

### Concerns on the results
- In table 1. The quantitively improvement is marginal in many metrics. It is not clear, how it translates to the qualitative results. E.g. 0.965 vs0.966 or 0.0308 vs 0.0374, 30.91 vs 31.10
- The visual results in Fig 4 , 3DGS-avatar seems very close to ToMiE than GART, but in table 1 quantitatively it is overall the inverse of it. Why is this ?
- In Figure 5, the results in the last two row, GART results seems same or better. What makes GART better?. Is this a general phenomenon that ToMIE performs better when there are hand-held object ?.
- What about non-rigid objects and different fabrics of garments. How to handle them in the context of animation?.
- The top and the bottom garments, are both of them considered two separate geometries/meshes or they are single. How to handle collision between these two?.
- How to handle garment self-occlusions?
- What is the sensitivity of the method w.r.t errors in mask computation, the speed of animation etc.
- The results in the videos are mostly blurry and are not sharp, no clear cut boundary between hand, objects and garments. Please explain.

**Questions:**

Please see the weakness section.

---

### Note · Authors · 2024-11-13

I have read and agree with the venue's withdrawal policy on behalf of myself and my co-authors.